# Formation of Korean Christianity through the Banning of Ancestral Rites

**Shinhyung Seong** 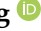

Baird College of General Education, Soongsil University, Seoul 06978, Republic of Korea; seongshil@ssu.ac.kr

**Abstract:** This study explores the ways in which a ban on ancestral rites influenced Korean Christianity. Ancestral rites are religious ceremonies that form the most critical social basis of Joseon, a Confucian society. First, the Korean Catholic Church was the first to oppose ancestral rites. Catholics created a new social and ethical resonance in Joseon society but had to endure tremendous persecution. Second, Protestantism was introduced when Joseon society was the most confused. Protestant missionaries banned ancestral rites, and Korean Protestants accepted them. Gradually, they interpreted it and embodied it in their faith. The ban on ancestral rites contributed to the formation of Korean Christianity. This laid the foundation for Christian social ethics and *Hyo* (孝, *Xiao* (Chinese pronunciation), filiality) theology. It has expanded into various fields, such as systematic theology, biblical studies, practical theology, and liturgical practice. Thus, this study examines how the ban on ancestral rites in Korea had a profound impact on the contextualization of Korean Christianity.

**Keywords:** ancestral rites; contextualization; Korean Christianity; *Hyo* (filiality) theology; liturgical practice

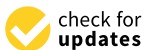



## 1. Introduction

Theology as a process of contextualization ([Bevans 2002](), p. 3) led to the formation of Christianity when the first Jewish disciples of Jesus transmitted their teachings to Greek culture. Christianity spread worldwide and developed to suit the circumstances of each region. Christianity has no unique theology but rather a contextual theology. Specifically, Christianity has evolved through the traditional confessional theologies in ways related to the enculturating process of the fundamental doctrines of Christianity, like Christology and the Trinity.

Accordingly, this study examines ancestral rites, one of the most critical issues in introducing and developing Christianity in Korea. It examines how Christianity, a Western religion, was contextualized in Korea, and how theology and liturgy were formed. First, it examines the story of the Korean Catholic Church's refusal to perform ancestral rites. In particular, the background and results are explored, focusing on the *Jin-san* incident, in which Catholics refused to perform ancestral rites for the first time. Second, this study examines how Korean Protestant churches rejected ancestral rites. In Korean Protestantism, the controversy over the rejection of ancestral rites developed as church members accepted and amplified missionaries' claims. Finally, this study examines how Christianity became indigenous to Korean society through this process from social–ethical, theological, and liturgical perspectives. This study examines one aspect of the formation of Korean Christianity.

## 2. Religious Meaning of Ancestral Rites of Confucianism and its Social Context in Korea

Ancestral rites are the main ceremonies that best reveal the religious characteristics of Confucianism. From the standpoint of viewing Confucianism as a form of religion rather than a political ideology, ancestral rites have been emphasized as the center of Confucian

tradition. Three foci are involved in interpreting ancestral rites as religious forms. First, people receive blessings from them. Second, ancestral rites make humans realize the extinction and existence of the soul. Third, ancestral rites help people realize the origin and the ultimate (Lee 2011, pp. 476–85; Bae 2013, pp. 419–23).

Confucianism emphasizes that those who perform ancestral rites are blessed. Confucius once said, "I receive blessings when I offer sacrifices (*The Book of Rites*, Chapter 10)."[1] Humans receive blessings from God through sacrifices and meet God through this process. There are seven main stages of ancestral rites:[2] humans welcome the gods (*yeong-sin*, 迎神), humans offer sacrifices to the gods (*jin-chan*, 進饌), humans raise drinks to the gods (*heon-jag*, 獻酌), gods accept humans' devotion and offerings (*heum-hyang*, 歆饗), gods respond to humans with blessings (*gang-bog*, 降福), humans receive and keep the blessings given (*eum-bog*, 飲福), and humans send gods with courtesy (*song-sin*, 送神).[3] Humans conduct the first, second, and third stages toward gods. Gods conduct the fourth act and the fifth stage to humans. The sixth and seventh stages involve humans' building relationships with gods. Among these, the fifth and sixth steps are directly related to blessings (Bae 2013, p. 419).

The blessings received through ancestral rites do not mean praying for materialistic blessings but are obtained through the mutual relationship between gods and humans. In other words, it is a blessing that prays for people to live well in the world through encounters between God and humans. Chapter 25 of *The Book of Rites* elaborates this point:

> The ancestral rite of a virtuous person is sure to receive the blessing. It is not the blessing that the world calls, but being equipped (*bi*, 備)). Being equipped means being compliant in all things (*sun*, 順)). It implies that he or she is equipped to follow the doctrine, which means that they dedicate themselves internally and follow the way (*do* (Tao, Chinese pronunciation) 道,) externally. 賢者之祭也, 必受其福, 非世所謂福也. 福者 備也, 百順之名也. 無所不順者謂之備, 言內盡於己, 而外順於道也. (The Book of Rites 2003, Chap. 25)

Thus, the blessings received through ancestral rites do not mean acquiring material belongings. However, since virtuous people already live according to the principles (the Tao, *do*, 道) of everything, this life is a blessing from the gods.

Moreover, ancestral rites help people understand the existential limits of life and death and accept the world after death. In other words, ancestral rites make people realize that the dead and the living interact even after death. Confucian ancestral rites divide the soul into *hon* (魂) and *baeg* (魄). *Hon* (魂) is the energy from heaven, and *baeg* (魄) is the shape of the earth. Heaven is related to the *yang* (陽, positive) principle, and the earth is related to the *eum* (陰, yin (Chinese pronunciation), negative) principle. In other words, when humans die, they return to the principles of *eum* (*yin*) and *yang*, the basis of harmony (Lee 2011, p. 484). In Confucianism, however, the soul does not have permanence but exists in the relationship of gathering *hon* and *baeg* at birth and dispersing them at death. "Dispersion" means that an individual disappears and returns to the universal energy of *eum* (*yin*) and *yang*. In Confucianism, the soul is said to remain intact for approximately 100 years (four generations) after death and then gradually disappears. Thus, ancestral rites were performed for about four generations (Lee 2011, p. 485). In the Confucian ancestral rite tradition, there is no separate mediator, such as a priest performing ancestral rites. The first-born son in a family performed the ceremony. Families are living beings whose ancestors and descendants are connected, and ancestral rites intertwine the living and dead. The person who conducts ancestral rites must live his or her life sincerely as a mediator connecting the living and dead (Lee 2011, pp. 486–87).

Lastly, the fact that death is not the end, realized through ancestral rites, naturally makes people aware of the existence of gods since they can reach the realm of ultimate origin through ancestral rites. Humans meet God through ancestral rites, such as blessings. Sincerity (*seong*, 誠) is needed for this process to occur. Ancestral rites are the foundation for reaching heaven wholeheartedly. The purpose of ancestral rites is to return to the original form. Ancestral rites are primarily to honor the parents who gave birth and, ultimately,

to honor heaven, the source of everything (Lee 2011, pp. 478–80). Confucianism eventually aims to reach heaven through self-discipline, that is, the process of uniting heaven and man. The reason for performing ancestral rites with sincerity is not to receive material blessings, appease the gods, or avoid their wrath and curses. The ultimate goal is to interact with and respond to God through ancestral rites (Bae 2013, pp. 422–23). Confucianism does not have the concept of monotheism like Western Christianity. It is difficult to understand the idea of God in Confucianism through Western concepts, but if we distinguish it, the idea of Confucianism is close to henotheism. There are three types of divine beings in Confucianism as the objects of ancestral rites: *Cheon-sin* (天神) (or *Sang-je*, 上帝, the god of the sky), *Ji-gi* (地祇, the god of the earth), and *In-gwi* (人鬼, ghosts of deceased ancestors). *Cheon-sin* is the God who presides over everything in the sky; *Ji-ji* refers to the mountains, rivers, and plants on the earth; and *In-gwi* is the god of the dead. The gods follow heaven's orders and interact with each other, and humans live with the gods to reach heaven (Bae 2013, pp. 408–10). Although Confucianism's view of God is far from Christian monotheism, it also clearly understands divine beings, as revealed through living in response to divine beings in specific situations.

During the Joseon Dynasty,[4] Confucian ancestral rites became deeply entrenched in the lives of Joseon people. The upper and lower classes regarded ancestral rites as significant ceremonies, even if the lower class could not prepare sufficient sacrifices. The desire to examine oneself and reach heaven and ancestral spirits became deeply rooted in people's lives during the Joseon Dynasty. The ancestral rites that played this role became a tool for creating social contradictions during the late Joseon Dynasty. Ancestral rites became a tool for building a social system that valued patriarchal, male-centered, primogenitary, and polygamous systems (Moon 1974, p. 75).

The reason for strengthening these social contradictions can be found in the code of clan regulations (*jong-beob*, 宗法) that has been strengthened since the 17th century. The code of clan regulations is a system of tribal organizations established during the Zhou Dynasty in China to establish a legitimate and primogeniture male-inheritance system. Family rules such as ancestral rites and marriage are regulated through this code. Neo-Confucian scholars accepted and established it in Korea at the end of the Goryeo Dynasty. In Joseon, this law was strengthened after the 17th century and became an important instrument for protecting social ideology. This code strengthened male-centered inheritance rights related to ancestral rites and women's subordination to men in the home (J. Kim 2002, pp. 38–9).

During this dynasty, women were subordinate to men. If the husband died during the early Joseon Dynasty, a woman could be listed as the head of the family (household); however, this could not be done during the late Joseon Dynasty. After her husband's death, a woman became dependent on her sons. Women were utterly subservient to men and were not given the right to handle matters based on their independent judgment. Women's rights and duties have disappeared, leaving only protection and exclusion. This relationship was also revealed in ancestral rites. In the early Joseon Dynasty, a woman had the authority as the daughter-in-law of the head of the family, known as the right of the head of the family (*chong-bu-gwon*, 冢婦權). If the husband died childless, the woman (*chong-bu*, 冢婦) had the right to take her husband's place in ancestral rites. However, during the late Joseon Dynasty, this authority disappeared. When the day of ancestral rites comes, it is rare for a woman to participate in ancestral rites, and she is relegated to being a person who only prepares food and other materials for ancestral rites (S. Kim 2016, pp. 31–3).

In this situation, the Korean people accepted Christianity as a Western religion. They accepted the Catholic Church first in the 1870's. And they accepted the Protestant Church in the 1890's. The teachings of Christianity as a foreign religion posed the social framework of the Joseon Dynasty related to the practice of ancestral rites.

### 3. Korean Catholic Church and Ancestral Rites

The introduction of Christianity, a Western religion, into Joseon was an extraordinary phenomenon. Catholic missionaries like Francis Xavier and Matteo Ricci consecutively arrived in East Asia, such as Japan in 1549 and China in 1601. Catholic missionaries may have initiated the Catholic churches in East Asia. Meanwhile, there is a theory about the origin of *Im-jin-wae-lan* (the Japanese Invasion of Korea in 1592) that Céspedes, a Catholic missionary who was active in Japan with the Japanese army, came to Joseon in 1593 and first spread the Catholic Church (Kim et al. 2009, pp. 107–16). However, this theory is not currently accepted. Instead, after the *Byeong-ja-ho-lan* (the war between the Qing Dynasty (China) and the Joseon Dynasty, December 1636–January 1637), the Joseon Dynasty learned about the Catholic Church in China through the *Buyeon* envoy (*bu-yeon-sa-haeng*, 赴燕使行) to China through the travel of the noble class, known as *yeon-haeng* (燕行) in Korea. Joseon people were able to access Western books. These books were mainly Catholic Church books (Cho 2006, p. 200). Joseon people, especially those from the noble class who came and went as envoys, naturally accepted Western books as Western Studies (*Seo-hag*, 西學). Some gradually converted to the Catholic Church. This was the first time in the history of the World Church that Christianity was accepted voluntarily but not by missionaries.

In the late Joseon Dynasty, Joseon society divided ideological systems into right ideology (*jeong-hag*, 正學), practical ideology (*sil-hag*, 實學), and pseudo-ideology (*sa-hag*, 邪學). *Jeong-hag* was Neo-Confucianism, and *sil-hag* was a practical study to reform the theoretical rigidity of Neo-Confucianism. Buddhism and folk religion were in the category of *sa-hag*, and Western learning (*Seo-hag*, 西學) was newly added. *Jeong-hag* and *sil-hag* were socially recognized, but *sa-hag* was not publicly accepted (Cho 2003, pp. 51–2). The acceptance of the Catholic Church by the Joseon people, which began voluntarily with the introduction of Western learning (*Seo-hag*), reflected the social phenomenon of that time, including the demand for a changing social structure and the acceptance of advanced Western culture, that is, modernization. This was a result of a combination of these demands (K. Cho 2003, pp. 53–4; J. Kim 2002, p. 9).

Lee Seung-hun, the first baptized Catholic Christian in Korea, founded the first Catholic Church in Joseon in 1784, known as the initiating year of the Korean Catholic Church. He was a nobleman, received baptism by Missionary Jean-Joseph de Grammont in the Qing Dynasty, and returned to establish and spread the Catholic Church. Lee became Catholic at the recommendation of Lee Byeok, who had already become Catholic. He went to the Qing Dynasty, received the world, and returned to Joseon to establish a church. In areas without priests, a baptized person could build a church and officiate at mass in the Catholic tradition. Lee established the first Catholic Church as a substitute priest (Grayson 2005, p. 9). The Catholic Church began to spread among the noble class, gradually spread to the middle class, and was accepted by commoners. It spread quickly among those who wanted to reform the social hierarchy of the time:

> While the church was established by the scholar-gentry class, the first leaders included members from the middle people and commoners. An early Catholic adherent, the butcher Hwang Ilgwang, was of the lowest class and hence despised in society. When he became a member of the Catholic community, he was overwhelmed with emotion for being treated as a peer…… Other members of community released their slaves. (Yoon 2007, p. 357)

They also translated and read the Bible (the Gospel of Jesus) into Hangul, the Korean alphabet created by King Se-jong but ignored by the upper class because they respected only Chinese culture, preached the Gospel to women, and created a new community (J. Kim 2002, pp. 12–3; Yoon 2007, pp. 357–58).

The Catholic Church quickly took root in Joseon society but soon suffered persecution. The Catholic Church, which emphasizes monotheism, banned ancestral rites, and Catholic believers in Joseon followed this order. This belief became a leading cause of persecution. The trend of rejecting ancestral rites should not be understood simply at the religious level; rather, the desire for social change at that time should be understood. As mentioned ear-

lier, ancestral rites had great religious significance, and when performed correctly, they were a system that could establish relationships with gods and people. However, this spirit was not properly implemented in society, and it became a system that caused social discrimination. The religious reason for Catholics to reject ancestor rites was to abandon their polytheistic worldview and worship one God; however, this went beyond a simple religious trend and developed into a trend of social change. Accordingly, those who had a vested interest in Joseon persecuted Catholics severely.[5]

The Jinsan Incident, which occurred in Jinsan, Jeolla-do, in 1791, was the first incident in which the Catholic Church was persecuted for refusing to perform ancestral rites. This incident occurred when the Catholics Yoon Ji-chung and Kwon Sang-yeon refused to perform ancestral rites and burned the memorial tablet that had been used to enshrine their parents. These two people were cousins and, after becoming Catholics, refused to perform ancestral rites in accordance with the Vatican (Clement XI) order, banning ancestral rites in 1790. They destroyed all the ancestors' memorial tablets. When Yun Ji-chung's mother passed away, they refused to conduct ancestral rites but held the funeral following the traditions of the Catholic Church. Yun Ji-chung stated several reasons for refusing ancestral rites: The Church prohibits ancestral rites; since people go to heaven or hell when they die, it is unnecessary to enshrine their souls in a shrine; even if alcohol and food are offered to ancestors, the souls of the dead cannot eat; and filial piety (or filiality, *hyo*, (Xiao) 孝) is not about ancestral rites but about accumulating virtue (H. Cho 2018, pp. 149–50). Ultimately, these two people were sentenced to death for breaking the laws of the country.

Another thing to pay close attention to in this case is Yoon Ji-chung's mother. Unfortunately, her name is unknown, but she is mentioned as Madam of Andong Kwon Family in the dictionary, which also reflects the situation of that period. Mrs. Kwon lost her husband at a young age and raised her son, Yoon Ji-chung, and her nephew, Kwon Sang-yeon. She followed the Catholic Church's order banning ancestral rites as soon as they were issued in 1790. For women, following these orders was a case of pursuing human relationships that were completely different from the norms of society. As mentioned earlier, women in the late Joseon Dynasty were victims of the patriarchal system and were treated as shadows, given only their duties but no rights. However, before her death, she left a will not to hold Confucius' ancestral rite for her but a Catholic funeral service (S. Kim 2016, pp. 34–7). Yoon Ji-chung followed his mother's will and also endured martyrdom along with Sang-yeon Kwon, with whom he grew up. This example shows that the influence of the Catholic Church on Joseon society at the time was not merely religious but also social.

In this way, the Catholic Church in Joseon society underwent a trend of change in the modern sense; at its center was the rejection of ancestral rites, which had received considerable attention. This issue led to constant persecution, which became a good excuse for persecution. Meanwhile, the family was divided into those with and without faith, which also caused conflict within the family. Accordingly, in 1939, the Catholic Church (Pope. Pius XII) partially permitted ancestral rites as it banned serving ancestral tablets.

## 4. Korean Protestant Church and Ancestral Rites

Before understanding the origins of Korean Protestantism, it is necessary to examine the circumstances of the time. Protestantism was introduced to Korea in 1884 with the arrival of the medical missionary Allen and in 1885 when the missionaries Appenzeller and Underwood arrived. During this period, Joseon became politically and economically impoverished as the maternal relatives of the king became the ruling power with the fall of the royal authority. Along with the development of the Catholic Church, which started with *Seo-hag*, a new religion named *Dong-hag* (Eastern learning, 東學) emerged after Choi Je-woo received revelation from *Sang-je* (God) in 1859. Although there is a strong connection between *Dong-hag* and *Seo-hag* (K. Cho 2003, p. 52), *Dong-hag* developed religious ideas by combining traditional Korean Confucianism, Buddhism, and Taoism, and spread rapidly among the people, even sparking a revolution in 1894. The Joseon Dynasty suppressed the *Dong-hag* Revolutionary Forces with the help of the Qing Dynasty and Japan.

Regarding international relations, Joseon, whose power was weak, could barely maintain its national power among its neighboring countries, China (Qing), Japan, and Russia, along with the expansion of imperialism. Japan—which concluded a unilateral unequal trade treaty named the Gang-hwa-do (Gang-hwa Island) Treaty in 1876—strengthened its power on the Korean Peninsula, eventually led the war against the Qing Dynasty to victory (1894), and won the war with Russia in 1904. Finally, Japan became the most potent imperial power in the Korean Peninsula. During these chaotic times, Protestant missionaries came to the Korean Peninsula.

Many missionaries who came to Korea were American missionaries passionate about evangelizing the Gospel due to the influence of the revival movement that developed after the American Civil War in 1865. They emphasized pietism, Bible-centeredness, and social concerns (Grayson 2007, p. 434).[6] Although some Protestant missionaries, such as Hulbert and Gifford, cooperated with the Catholic Church, most carried out missionary activities while differentiating themselves from the Catholic Church. They thought that Korea was a wasteland for religion because of Koreans' multi-layered religious views. However, they gradually came to understand the meaning of religion by understanding the monotheistic faith of the *Dan-gun* (檀君)[7] myth that Korea had, and by experiencing the social revolution (1894) and revival (1907), they began to understand the religious nature of Korea (Oak 2013, pp. 66–83).

Because the circumstances of the times had already changed significantly, the issue of ancestral rites was not as central as it was in the Catholic Church. However, the issue of ancestral rites has also been dealt with in depth in Protestant churches. First, pamphlets written by Chinese Protestant missionaries Medhurst and Nevius were widely read in Joseon and greatly influenced the people. In particular, Nevius' *Errors of Ancestor* Worship (*Saseon-byeon-lyu*, 1864) became a guidebook for missionaries in Joseon. These two missionaries represented the first and second generations of Protestant missionaries in China, and during this period, there was no significant opposition to opinions refusing ancestral rites. However, as progressive missionaries entered China and a trend of positive evaluation of Chinese culture was formed, three missionary conferences were held in Beijing, China in 1877, 1890, and 1907, discussing banning ancestral rites. Rather than having a positive opinion on ancestral rites, the conclusion favored banning them (Oak 2013, pp. 357–74).

Protestant missionaries in Korea followed the opinions of Chinese missionaries from the beginning of their mission in Korea. The Korean Presbyterian Churches and Methodist missionaries, including Underwood, Appenzeller, Ross, and Scranton, implemented a policy of banning ancestral rites for several reasons. First, ancestral rites constitute idolatry and violate the first and second commandments of the Decalogue. Second, ancestral rites teach the soul immortality through unbiblical teaching. Third, in terms of social ethical content, ancestral rites are the direct cause of Korea's evil customs, such as early marriage, the concubine system, discrimination against women, and poverty. Fourth, as they tried to differentiate themselves from the Catholic Church, they argued that ancestor worship was a variation of the Catholic Church's saint worship and purgatory theory (Oak 2013, pp. 375–77). For these reasons, early Protestant missionaries in Korea opposed ancestral rites. Joseon people generally followed this view. Even when missionaries like Gale took a reserved stance by presenting cautious views, Korean Protestants strongly opposed ancestral rites. In September 1904, a missionary conference commemorating the 20th anniversary of the Korean mission was held in Seoul. Engel, an Australian Presbyterian missionary, expressed caution regarding this issue, and Moose agreed with this opinion; however, many other missionaries refused to perform ancestral rites (Oak 2013, pp. 283–85).

Interestingly, both the missionaries and Korean Christians participated in the debate by presenting their opinions on the issue of ancestral rites. Previously, there was a high conversion rate to Protestantism among lower-class grassroots groups below the middle class, including merchants and technicians. However, after the Russo-Japanese War in 1904, many members of the noble class entered the Protestant Church (Oak 2013, p. 382).

The debate became more active as Joseon Protestants contributed articles to new media outlets such as the *Christ Newspaper* (*Geu-li-seu-do-sin-mun*) (1897) and *Joseon Christian Bulletin* (*Jo-seon(dae-han)-keu-li-seu-do-in-hoe-bo*) (1899), and missionaries expressed their opinions there (Oak 2013, p. 393).[8]

The debate progressed as Joseon Christians who were well versed in Confucianism expressed their opinions. In addition to the prevailing opinion that ancestral rites were idolated, Korean Protestants expressed their opinions on ancestral rites. Noh Byeong-seon and Choi Byeong-hun attempted a new interpretation by combining East Asian traditions with soteriology. Noh argued for open soteriology in his book *Pa-heug-jin-seon-lon*, published for Christian apologetics. It is a similar claim to Karl Rahner's Anonymous Christian, which claims that those who have not heard the Gospel are saved through their good deeds and merits (Noh 1899, p. 8b). Choi argued for a change from ancestral rites to Christian worship in his article "Jung-chu-ga-jeon-il,"[9] in the 27 September 1899 issue of *Joseon Christian Bulletin* (*Joy-seon(dae-han)-keu-li-seu-do-in-hoe-bo*), arguing that the old Confucian-style ancestral rites should be abolished, and ancestral rites should be completed through Christian worship (Choi 1899).

Thus, Korean Protestants developed their own theological and liturgical traditions regarding funeral culture while adhering to the ban on ancestral rites as they accepted missionaries' opinions about them. While the Catholic Church changed this policy from prohibition to permission, Protestants made efforts to create their traditions, and these efforts developed into a process through which Korean Protestants found their path.

## 5. The Formation of Korean Christianity through the Issue of Ancestral Rites

Christianity encountered varied cultures across countries in the field of missions, accepted and transformed each other, and sought theoretical foundation and practical measures. Just as early Christianity was contextualized by encountering Greek–Roman culture, in Korea, it was contextualized by encountering Confucian culture. The center of this phenomenon was the issue of ancestral rites. On the Korean Peninsula, Buddhist culture continued for approximately 1000 years throughout the Silla and Goryeo periods. However, when religion colluded with power and became corrupted, Neo-Confucian scholars opposed it and founded Joseon. Confucianism was a political philosophy, but it also played a religious role, with ancestral rites at its center. The ritual of observing oneself, reaching the ultimate goal, and meeting God while enjoying blessings through ancestral rites became the center of Joseon people's lives. However, after the 17th century of the Joseon Dynasty, the spirit disappeared, and ancestral rites became one of the essential elements causing social contradictions and conflicts. In this situation, Christianity (Catholicism first) was introduced into the Joseon Dynasty, and the Christian spirit opposing ancestral rites spread among Catholic Christians.

Christianity in Korea was formed and took root in response to ancestral rites that Confucianism valued the most in Joseon society. This is a potent social and ethical response. The Catholic Church was involved at the beginning of this process. In the late 16th and early 17th centuries, Joseon endured two invasions from Japan (1592–1598) and China (1627 and 1636).[10] Joseon society accepted Western learning (*Seo-hag*) while pursuing practical values to overcome wars' predicaments and take a new leap forward. Because the texts of *Seo-hag* were books from the Catholic Church, Joseon naturally accepted the Catholic Church. It opposed ancestral rites and rejected idol worship at a religious level. However, the problem was complex. At that time, ancestral rites were at the center of creating a Neo-Confucian family based on Confucianism. Because healthy social customs during the early Joseon period were broken by war, ancestral rites were used to discriminate against women and create a hierarchical structure centered on men, especially the eldest son. During this process, the social and ethical ramifications of the Catholic Church, which rejected ancestral rites while spreading universal love, were significant. They demanded human dignity and created a new community based on gender equality (J. Kim

2002, pp. 42–9; S. Kim 2016, pp. 39–40). Great persecution awaited Catholics; however, the newly ethically armed Catholic Church grew.

Meanwhile, Protestantism was introduced quite differently from the Catholic Church when Joseon was nearly collapsing. Protestant missionaries recognized various ethical problems in Korean society and confirmed that these problems were not separate from the issue of ancestral rites. Therefore, ancestral rites were prohibited not only from a religious perspective but also from an ethical perspective (Oak 2013, p. 356). Thus, in the formation of Korean Christianity, the issue of banning ancestral rites was connected to social situations. It had become a channel for presenting alternatives at the social and ethical levels.

Moreover, the ban on ancestral rites caused great controversy in Joseon society, which concerned filial piety (filiality, *hyo* (Xiao) 孝), one of the fundamental values of Confucianism. Refusing to perform ancestral rites is the greatest act of disloyalty toward parents. The Catholic Church responded to these claims with a readiness to die, that is, martyrdom. After the Jinsan Incident in 1791, more than 10,000 believers became martyrs in the persecution, which lasted for 100 years. This persecution ended in 1939 after ancestral rites were understood and accepted as part of traditional culture. At the same time, Protestantism developed the theology of filial piety. Korean Protestantism accepted the Confucian ethics of filial piety (filiality, *hyo* (Xiao) 孝), as Christian ethics and developed logic at the theological level. Protestants actively defended the fifth commandment of the Decalogue, arguing that serving one's heavenly father is the basis for honoring one's parents, and emphasizing that one should perform filial piety. Instead of performing ancestral rites after their death, people should conduct their filial piety while their parents are still alive (Oak 2013, pp. 401–3). Several Korean Protestant theologians are currently developing *Hoy* theology, which explores the meaning of filial piety at systematic theological, biblical, and practical theological levels (Ban 2011; Nam 2008; Oak 2013, pp. 419–20).

These theological endeavors eventually led to the development of liturgical practices such as Korean-style funerals and memorial services. In the Korean Catholic Church, it is difficult to find a developmental process at the liturgical level because ceremonies are determined according to the guidance of the Vatican. As ancestral rites were banned, using ancestral tablets and eating ceremonial food were prohibited. Additionally, ancestral rites for Confucius, the founder of Confucianism, maintained as a Confucian tradition, were also prohibited. After ancestral rites were allowed in 1939, however, only prohibitions on using ancestral tablets remained. Meanwhile, Protestantism's sacrificial ritual was called a *Chu-do-ye-bae* (追悼禮拜, memorial service) and became Christian content. "Prayers for the dead," practiced not only in Confucian ancestral rites but also in the Catholic Church, were banned, and non-Christian elements such as *Chug*-mun (祝文), *Heum-hyang* (歆饗), *Eum-bog* (飲福), and *Pung-su* (風水) were removed.[11] Instead, they invited church members to hold a simple worship service with hymns, prayers, and Bible reading and then changed the format to sharing food (Park 2012, pp. 128–9). Each Korean Protestant denomination creates and implements an order for memorial services (Park 2012, p. 133). This liturgical practice results from the development of *Hyo* theology in Korea. Thus, Confucian ancestral rites influenced Korean Christianity.

## 6. Conclusions

This study explored the issue of the ban on ancestral rites to examine the formation of Korean Christianity. Ancestral rites were religious ceremonies that served as the most critical social basis of Joseon, a Confucian society. The Korean Catholic Church was the first to oppose ancestral rites. As a result, it created a new social and ethical resonance in Joseon society. However, because this shook the foundations of society, Catholic Christians endured tremendous persecution. Protestantism was introduced when Joseon society was the most confused. Protestant missionaries, like the Catholic Church, banned ancestral rites. Korean Protestants experienced the process of the missionaries' ban on ancestral rites, as Protestants in Joseon interpreted and accepted it and embodied it in their faith.

Overall, the ban on ancestral rites became the most essential foundation for the formation of Korean Christianity. This laid the foundations for Christian social ethics. In particular, in conjunction with social changes in the late Joseon Dynasty, the rejection of ancestral rites brought about social changes at an ethical level. In addition, while interpreting filial piety from a theological perspective, it expanded into various fields such as systematic theology, biblical studies, practical theology, and liturgical practice, becoming the foundation for the formation of today's Korean Christianity. Thus, this study examines how the ban on ancestral rites in Korea had a profound impact on the contextualization of Korean Christianity.

**Funding:** This work was supported by the Ministry of Education of the Republic of Korea and the National Research Foundation of Korea (NRF-2018S1A6A3A01042723).

**Data Availability Statement:** No new data were created or analyzed in this study. Data sharing is not applicable to this article.

**Conflicts of Interest:** The author declares no conflict of interest.

## Notes

[1] *The Book of Rites* is one of the Five Classics of Confucianism. It is a compilation of Confucius and his disciples' writings on ancient Chinese etiquette, collected by Confucian scholars during the *Han* Dynasty. The Five Classics include *Classic of Poetry*, *the Book of Documents*, *the Book of Rites*, *I Ching* (*the Book of Changes*), and *Spring and Autumn Annals*. James Legge translated *the Book of Rites* into English in 1885 for the first time. In this paper, I followed the Korean translation published by Lee Sang-ok at *Myeongmundang* in 2003 and cited chapters instead of pages according to the usual clause notation in Korea.

[2] The ancestral rites have a preparatory stage before the main stage and a final stage afterward. The preparation stage is where the body is thoroughly bathed and cleaned (*je-gye*, 齊戒) and prepared foods (*jin-seol*, 陳設) to be offered to the ancestral spirits for ten days. In the final process, there is a stage of placing the sacred tablets in the shrine (*nab-ju*, 納主) and removing the offerings (*cheol-jo*, 徹俎), a stage of burning the offerings (*mang-lyo*, 望燎) and burying the ashes (*ye-mae*, 瘞埋) to make gods return after the ancestral rite, and a stage in which the ancestral rite participants share the foods (*bun-jun*, 分餕) to share the blessings from the gods (Bae 2013, pp. 416–9).

[3] Chinese characters originated in ancient China and are currently used in East Asian countries such as China, Japan, Korea, Vietnam, Singapore, and Malaysia. The Chinese characters in these countries are similar but used differently in their pronunciation and usage. Chinese characters are an element of the Korean language, and when Koreans read Chinese classics, they read them with Korean pronunciation, not Chinese. Therefore, since this paper describes Korean Christianity, it is written according to Korean pronunciation rather than Chinese. This paper used Korean pronunciation and followed the way of the Revised Romanization of Korea, which was released to the public on 7 July 2000, by the Ministry of Culture and Tourism. Chinese pronunciations widely used in academia are also indicated.

[4] Joseon (1392–1910) was a country founded by Confucian scholars (Neo-Confucians) who wanted to reform Goryeo (918–1392), which had Buddhism as its political and religious foundation. Accordingly, Confucianism established itself as a powerful ruling ideology during the Joseon Dynasty and exerted a strong religious influence through ancestral rites. However, the form of religious life during the Joseon Dynasty remained consistent with the form of Buddhism combined with shamanism for a long time. It is reasonable to say that Confucianism, the ruling ideology, simultaneously acquired a religious aspect, but Buddhism was the foundation of the people's religion.

[5] After entering Joseon, Catholics were severely persecuted four times: the *Shin-yu* Persecution in 1801, the *Gi-hae* Persecution in 1839, the *Byeong-o* Persecution in 1846, and the *Byeong-in* Persecution in 1866. Of course, this means that there were only four major bloody outbreaks, and in the meantime, large and small persecutions and executions continued continuously. In particular, the *Byeong-in* Persecution of 1866 was so harsh that it is estimated that at least 8000 to up to 20,000 people were martyred during this time alone.

[6] For a long time in Korean church history, the early missionaries were evaluated as fundamentalists, but viewing them as fundamentalists did not sufficiently consider the historical situation. This is because fundamentalism began in the United States after the 1930s. Regarding this, Oak Sung-Deuk said that the reason for the prejudice that the early missionaries in Korea were fundamentalists was that Korean historians followed the missionary report of Missionary Brown, who was the general secretary of the Overseas Mission of the Northern Presbyterian Church in the United States, where he introduced them as stubborn and exclusive, and Korean historians followed this idea. Ok pointed out furthermore that this was due to the excessive dichotomous thinking of progressive theologians in evaluating them as fundamentalists. However, they could be evaluated as people who showed their passion for the Gospel rather than fundamentalists (Oak 2013, pp. 45–55).

[7] Koreans have revered him as the founder of Korea. According to the myth of *Dan-gun*, he was born between *Hwan-ung* (a son of God, *Hwan-in*) and *Ungnyeo* (she was initially a bear, but she changed a woman after passing a test, enduring the ordeal of eating

only mugwort and garlic in a cave without seeing sunlight for 21 days), established his capital at *Asadal* in 2333 BC, established Gojoseon, and ruled the country for about 2000 years.

8   Underwood published *Christ Newspaper* (*Geu-li-seu-do-sin-mun*) in 1897, and Appenzeller published *Joseon Christian Bulletin* (*Joy-seon(dae-han)-keu-li-seu-do-in-hoe-bo*) in 1899; it was named *Jo-seon* first, but he changed it to *Dae-han*.

9   Jung-chu-ga-jeon-il, known as *Chu-seog* (秋夕), is a national holiday celebrated on the eighth month of the lunar calendar (15 August) in East Asia, similar to Thanksgiving Day in the U.S.A.

10  The Japanese invasion in 1592 is called the Im-jin-wae-lan. Meanwhile, the Yeo-jin tribe in China invaded Joseon for the first time in 1627, before the establishment of the Qing Dynasty (Jeong-myo-ho-lan). After establishing the Qing Dynasty, they invaded Joseon again in 1636 (Byeon-gja-ho-lan).

11  *Chug-mun* is written material to offer blessings to God, *Heum-hyang* is that gods accept humans' devotion and offerings, *Eum-bog* is that humans receive and keep the blessings given, and *Pung-su* is about finding good land (grave site).

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
