# Peer review of "Formation of Korean Christianity through the Banning of Ancestral Rites"

_religions, doi:10.3390/rel15030280_

Round 1

Reviewer 1 Report

Comments and Suggestions for Authors

The article is very interesting and written correctly. It fits into the study of the inculturation of Christianity.

I have no fundamental comments on the text.

Only in line 25-26 I suggest an addition - the author writes: "Christianity has no unique theology but rather a contextual theology". I would clarify what "unique" means. In its fundamentals, Christianity is consistent (the truth of the Trinity, Jesus Christ, etc.). It is worth pointing out that we can speak of confessional theologies. It is also important to clarify the contextual nature of theology - it is about the ability of Christianity to inculturate, to express the truth in the context of a given culture, but with the integrity of the fundamentals.

Author Response

Thank you for reviewing my article. I modified it according to your advice. 

Reviewer 2 Report

Comments and Suggestions for Authors

            The overall arguments are poorly grounded in evidence and too conclusive without being backed up by examples. The author suggests that in Korea Confucian ancestral rituals were incorporated into Christian social ethics, gave birth to Hyo-theology, and affected systematic theology, biblical studies, liturgical practice, and so on, but other than stating these conclusive remarks, no specific examples are discussed or provided in detail.

            The author’s understanding of Confucian ideas of filial piety, as well as of concomitant Confucian ancestral rites is too simple, too naïve, and too definitive and, as a result, preempts any balanced discussion of their rich traditions and variations and contradictions. The author seems to have a singular thread of understanding that only serves his/her own predetermined arguments. In particular, the term “God” is used in describing Confucian ideas of souls and ancestral rituals – a term that would frighten readers. The author seems to frame Confucian ideas and values within the scope of Christianity.

            Similarly, the author’s discussion of the history of Christianity in Joseon Korea is full of misunderstandings and wrong information. It is problematic. The author’s understanding of Korean traditional culture and society is also substandard. The author’s description does not reflect the current level of research in the field that deserves appreciation and should be reflected in the article.

Comments on the Quality of English Language

The Romanization of Korean terms is substandard. It seems that the author does not know how to Romanize them. Many alien terms are used without English translation and thus would trouble non-Korean speakers.

Author Response

Thank you for your review. As you advised, I tried to review and revise the Confucian tradition once again. In particular, I added more information on the Confucian idea about God in lines 108-111. Historical facts were also checked and corrected. Content related to the Catholic Church was added in lines 162-166 and was clearly stated in line 187. In addition, endnotes 3 and 7 explain information on the cultural background of the Korean language related to Chinese characters and Korean national mythology. Lastly, the English was refined once again through a professional editor (editage.com). I re-checked and corrected the Romanization, which was your advice, and added an explanation. Thank you again because your review helped me improve my writing.

Round 2

Reviewer 2 Report

Comments and Suggestions for Authors

Nothing to add

Comments on the Quality of English Language

Nothing to add